# Narrowband room temperature phosphorescence of closed-loop molecules through the multiple resonance effect

Xiaokang Yao[1,2,5], Yuxin Li[1,5], Huifang Shi [1,5], Ze Yu[1], Beishen Wu[1], Zixing Zhou[2], Chifeng Zhou[1], Xifang Zheng[1], Mengting Tang[1], Xiao Wang [2], Huili Ma[1], Zhengong Meng[1], Wei Huang[1,2,3,4] & Zhongfu An [1,2,4] ✉

Luminescent materials with narrowband emission show great potential for diverse applications in optoelectronics. Purely organic phosphors with room-temperature phosphorescence (RTP) have made significant success in rationally manipulating quantum efficiency, lifetimes, and colour gamut in the past years, but there is limited attention on the purity of the RTP colours. Herein we report a series of closed-loop molecules with narrowband phosphorescence by multiple resonance effect, which significantly improves the colour purity of RTP. Phosphors show narrowband phosphorescence with full width at half maxima (FWHM) of 30 nm after doping into a rigid benzophenone matrix under ambient conditions, of which the RTP efficiency reaches 51.8%. At 77 K, the FWHM of phosphorescence is only 11 nm. Meanwhile, the colour of narrowband RTP can be tuned from sky blue to green with the modification of methyl groups. Additionally, the potential applications in X-ray imaging and display are demonstrated. This work not only outlines a design principle for developing narrowband RTP materials but also makes a major step forward extending the potential applications of narrowband luminescent materials in optoelectronics.

Room-temperature phosphorescence (RTP) has aroused considerable attention in the field of organic light-emitting diodes[1,2], bio-imaging[3], anti-counterfeiting[4] and so on[5–7], because of its high exciton utilisation, large Stokes shift, long-lived emission lifetimes, etc. For purely organic matters, RTP is too weak to be detected under ambient conditions, owing to weak spin-orbit coupling (SOC) between the excited singlet and triplet states, strong non-radiative transitions of excitons from the excited triplet to ground states, and intense quenching from external quenchers[8,9]. During the past decade, great effort has been devoted to obtaining high-performance RTP by promoting the SOC process with the introduction of heavy atoms[10–13], carbonyl group[14], twisted

molecular structures[15], and suppressing non-radiative transitions with a rigid molecular environment construction by crystal engineering[16–20], polymerisation[3,21], supramolecular self-assembly[22,23], host–guest doping[24–27] and so forth. To date, a great success has been achieved in improving quantum efficiency, tuning phosphorescence lifetimes and regulating colour gamut. However, there is no report on the colour purity of phosphorescence, although phosphorescence colours can be controllably tuned from blue to red, even to near-infrared regions.

It is worth noting that the narrowband luminescence materials have a great advantage of high colour purity, which exhibits potential applications in high-resolution imaging[28], ultrahigh-definition

[1]Key Laboratory of Flexible Electronics (KLoFE) & Institute of Advanced Materials (IAM), School of Flexible Electronics (Future Technologies), Nanjing Tech University (NanjingTech), Nanjing, China. [2]The Institute of Flexible Electronics (IFE, Future Technologies), Xiamen University, Xiamen, China. [3]Frontiers Science Center for Flexible Electronics (FSCFE), MIIT Key Laboratory of Flexible Electronics (KLoFE), Northwestern Polytechnical University, Xi'an, China. [4]Henan Institute of Flexible Electronics (HIFE) and School of Flexible Electronics (SoFE), Henan University, Zhengzhou, China. [5]These authors contributed equally: Xiaokang Yao, Yuxin Li, Huifang Shi. ✉e-mail: iamzfan@njtech.edu.cn

displays[29,30], sensing[31], and so on. So far, a series of narrowband luminescent materials have been developed with various approaches, including inorganic quantum dots[29], perovskite-based emitters[30] and organic dyes with rigid/fused aromatic structures[32,33], etc.[34,35]. For instance, a series of conventional fluorescent molecules, such as quinacridone and boron dipyrromethene dyes (BODIPY), are designed with bulky substituents to obtain narrowband emission[32,33]. Recently, the design principle of multiple resonance effects has also regulated the narrowband emission of thermally activated delayed fluorescence (TADF) molecules[36–41]. Researchers have yet to report a purely RTP molecule with a narrowband emission.

To date, the reported RTP materials exhibit poor colour purity with broad full width at half maxima (FWHM) exceeding 50 nm (Fig. 1a and Supplementary Note 1), which is ascribed to the intrinsic vibrionic coupling and structure relaxation of the π-conjugation molecules in the excited state[36]. Among them, the highest occupied and lowest unoccupied molecular orbitals (HOMO, LUMO) are localised between two atoms to form π-bonds (Fig. 1b). Therefore, there are stronger electron-vibration interactions for the transition from the excited triplet to ground state and the vibrational relaxation at $T_1$, thus resulting in broad phosphorescence emission for low colour purity (Fig. 1c). Inspired by the molecular design of the multiple resonance effect for narrowband fluorescence, we speculate that the multiple resonance molecules with some groups for the SOC promotion may be promising RTP candidates with high colour purity. As exemplified by a close-loop fragment of the quinolino [3,2,1-*de*]acridine-5,9-dione (Fig. 1b), HOMO and LUMO are localised at particular atoms owing to multiple resonance effect, which benefit to minimise the vibrionic coupling and structure relaxation for narrowband phosphorescence emission.

Because there is little change of nuclear shift between the excited triplet state and the ground state (Fig. 1c and Supplementary Note 2). Furthermore, the modification of the carbonyl group can improve the SOC to boost RTP behaviours. Therefore, we reason that molecules with such a rigid fragment may enable narrowband phosphorescence to enhance colour purity at room temperature.

To validate our hypothesis, a series of quinolino [3,2,1-*de*]acridine-5,9-dione derivatives, namely 7-methylquinolino[3,2,1-*de*]acridine-5,9-dione (7-MQ), 3-methylquinolino[3,2,1-*de*]acridine-5,9-dione (3-MQ), quinolino[3,2,1-*de*]acridine-5,9-dione (QA) and 3,7,11-trimethylquinolino[3,2,1-*de*]acridine-5,9-dione (TMQ) were designed and synthesised by the Ullmann and Friedel-Crafts acylation reactions with yields of 36.8%, 24.5%, 39.3% and 18.4%, respectively. The chemical structures of the target molecules were confirmed by nuclear magnetic resonance (NMR) (Supplementary Figs. 3–18). As anticipated, colourful narrowband phosphorescence was obtained when the phosphors were doped into a benzophenone (BP) matrix. The FWHM is as low as 30 nm at room temperature, and it is only 11 nm at 77 K. Besides UV light, the narrowband RTP can be also excited with X-ray. The potential applications in X-ray imaging and display were demonstrated based on 7-MQ. This finding will outline a facial principle of designing narrowband RTP molecules for various potential applications in the future.

## Results

### Photophysical properties of organic phosphors

Firstly, we selected 7-MQ as a model to investigate the photoluminescence (PL) properties of the target phosphors in dilute solution and solid state. In dilute 2-methyltetrahydrofuran (*m*-THF) solution ($1 \times 10^{-5}$ M) of the 7-MQ molecule, there is a broad

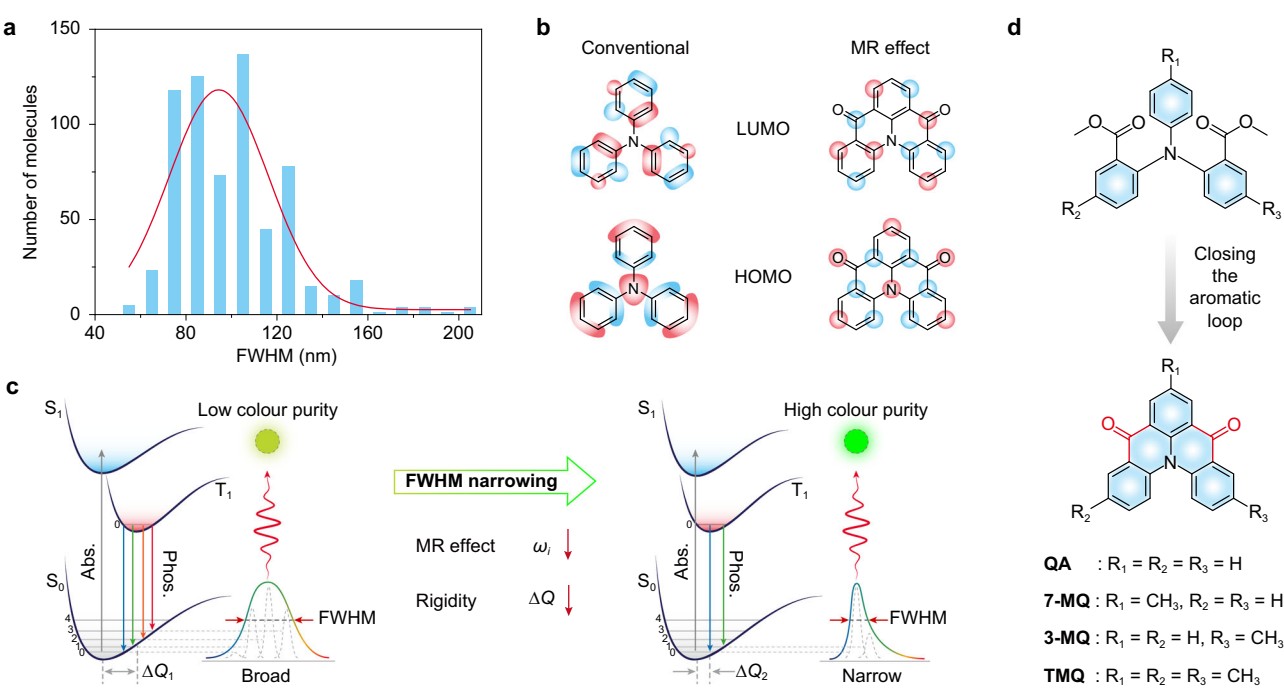

**Fig. 1 | Schematic illustration of the narrowband phosphorescence in purely organic phosphors. a** Statistical distribution of FWHM of the RTP spectra based on the reported RTP molecules. The blue bars and the red curve represent the number of molecules and the fitted curve, respectively. **b** HOMO and LUMO models of a conventional emitter of triphenylamine and a multiple resonance emitter of QA. **c** Proposed mechanism for manipulating FWHM of phosphorescence. According to the equation of Huang–Rhys factor ($S_i$), it is a way to obtain a small $S_i$ is that reducing the values of $\omega_i$ and $\Delta Q$ simultaneously. For traditional molecules, there exists broad FWHM owing to vigorous vibration and structure relaxation of the π-conjugation molecules from the excited state to the ground

state, leading to poor colour purity. After close loop of aromatic units, the molecular structure becomes more rigid, suppressing structural distortion, resulting in small $\Delta Q$. Meanwhile, the formation of MR effect by induction of the carbonyl groups can significantly suppress stretching vibrations, thus leading to a very low $\omega_i$. Therefore, the construction of rigid chromophore architectures with the MR effect can be an approach to obtain narrowband phosphorescence for high colour purity. Abs. and Phos. are the abbreviation of absorption and phosphorescence, respectively. **d** Chemical structures of the QA, 7-MQ, 3-MQ and TMQ phosphors. Source data are provided as a Source Data file.

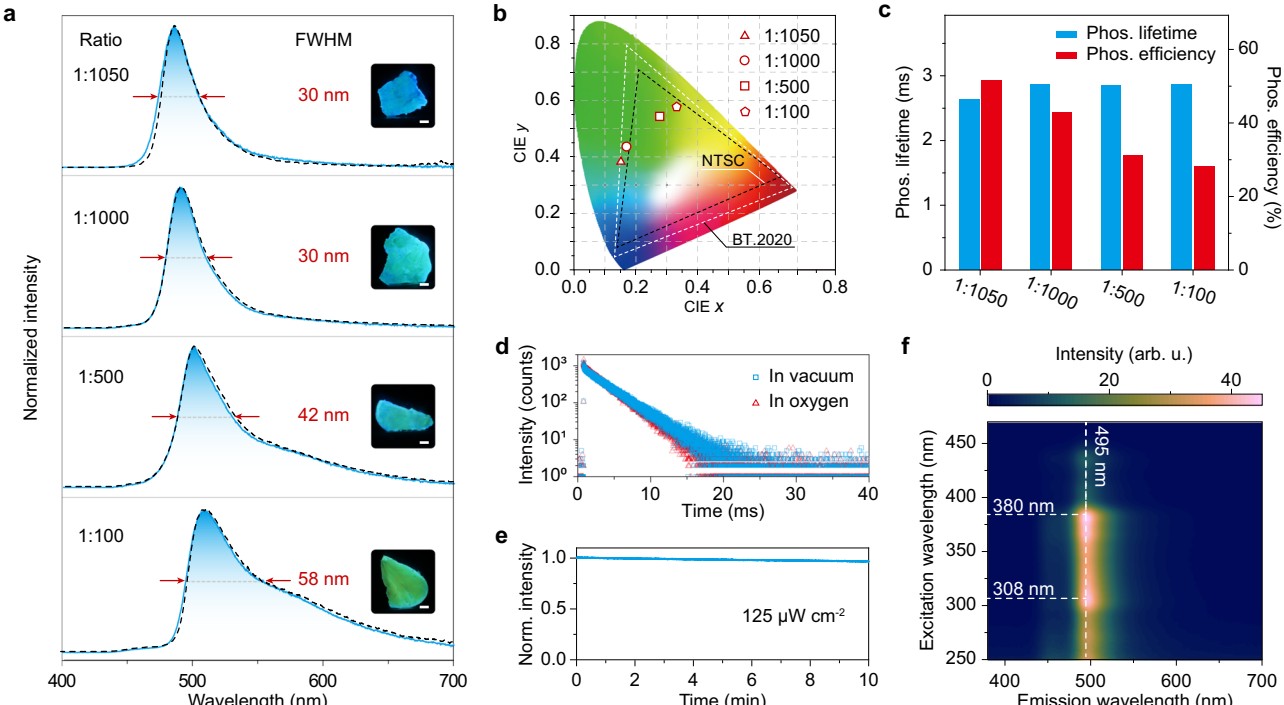

**Fig. 2 | Photophysical properties of 7-MQ doped into BP matrix (7-MQ/BP) under ambient conditions. a** Normalised steady PL (dotted lines) and delayed (solid lines) spectra of the 7-MQ/BP phosphors with various ratios. Insets are photographs of the phosphors taken under 380 nm UV light. The delay time is 8 ms. Arrows indicate FWHM. The scale bar is 1 mm. **b** The CIE coordinates diagram for the phosphorescence of the phosphors with various ratios. **c** phosphorescence lifetimes and efficiency of the 7-MQ/BP phosphors. **d** Phosphorescence decay curves of the 7-MQ/BP phosphor with a ratio of 1:1000 in vacuum and after exposure in oxygen atmosphere for 10 min. **e** Phosphorescence intensity at 495 nm of the phosphor with a ratio of 1:1000 as a function of time under excitation by 380 nm UV light with a power of 125 μW cm⁻². **f** Excitation-phosphorescence mapping of the 7-MQ/BP phosphor with a ratio of 1:1000. Source data are provided as a Source Data file.

absorption band from 350 to 470 nm (Supplementary Fig. 19), which is ascribed to intramolecular charge transfer transition[38]. Under UV-light irradiation, the solution shows blue emission with a peak at 464 nm at room temperature. The emission at 464 nm shows a life-time of 4.3 ns in *m*-THF solutions (Supplementary Fig. S20 and Supplementary Table 2), indicating the emission is fluorescence. As shown in Fig. 2a, the steady-state and delayed PL spectra of 7-MQ doped into BP matrix (7-MQ/BP) have a great overlap under ambient conditions. As the doping ratio of 7-MQ decreases from 1:100 to 1:1050, the FWHM of emission gradually narrows from 58 to 30 nm. It is the narrowest FWHM among the reported RTP molecules. The broad FWHM was ascribed to the aggregation of 7-MQ molecules as doping ratio of increase (Supplementary Fig. 21). Meanwhile, the emission profiles show blue-shifted with peaks changed from 509 to 485 nm, which corresponds with Commission International de l'Eclairage (CIE) of (0.333, 0.578), (0.278, 0.544), (0.171, 0.437) and (0.152, 0.383), respectively (Fig. 2b). With the ratio changes of the 7-MQ dopant, there is almost no change in the lifetimes of the phosphors (Fig. 2c). From Supplementary Fig. 22, it was found that the decay curves showed mono-exponential profiles with lifetimes of around 2.85 ms. We reasoned that the emission of the phosphors is phosphorescence under ambient conditions, which is further con-firmed by the following experiments. The phosphorescence effi-ciency of the phosphor reaches up to 51.8% when the doping ratio of 7-MQ is 1:1050 under ambient conditions.

In a further set of experiments, we investigated the phosphores-cence stability of the organic phosphors. As shown in Fig. 2d, the phosphorescence lifetimes show a certain degree of reduction from 3.2 to 2.5 ms when the phosphors change from a vacuum to an oxygen atmosphere. There is only a 3.4% reduction in the intensity of phos-phorescence emission at 495 nm under continuous irradiation for

10 minutes by 360 nm UV light with a power of 125 μW cm⁻² (Fig. 2e). In addition, 7-MQ/BP also exhibited excellent stability under N₂ and O₂ atmospheres (Supplementary Fig. 23). The limited quenching is ascri-bed to isolation protection by the rigid molecular environment of the crystalline matrix, which is confirmed by the powder X-ray diffraction (PXRD) (Supplementary Fig. 24). There exist similar PXRD patterns of the doping phosphors (7-MQ/BP) with multiple sharp peaks to BP crystal. With the excitation wavelength changed from 300 to 450 nm, there is also no change in the profiles of phosphorescence spec-tra (Fig. 2f).

## Investigation for narrowband RTP

To gain deeper insight into the efficient RTP performance of 7-MQ in the BP matrix, we first conducted a set of experiments on temperature-dependent PL, lifetimes, and excitation spectra. As shown in Fig. 3a, the steady-state photoluminescence intensity of the 7-MQ/BP phosphor exhibited a gradual decrease as the temperature increased from 100 to 300 K. The main emission bands with peaks at 495 nm show no var-iation across different temperatures. Similar to the photo-luminescence spectra, the temperature-dependent delayed spectra of the 7-MQ/BP also demonstrate a decreasing trend as the temperature increases to 300 K. (Supplementary Fig. 27). Meanwhile, it was observed that the emission lifetimes at 495 nm continuously decrease from 405 to 2.8 ms (Fig. 3b). However, the emission nature undergoes a transition from phosphorescence to thermally activated delayed fluorescence after doping the 7-MQ into the matrices of 9,9'-(1,3-phe-nylene)bis-9*H*-carbazole (mCP) and bis[2-(diphenylphosphino)phenyl] ether oxide (DPEPO) (Supplementary Figs. 28–31). Following sys-tematic analysis, we observed that the emission characteristics of the emitter are heavily influenced by host matrices[37]. Based on the afore-mentioned results, we concluded that the emission nature of the 7-

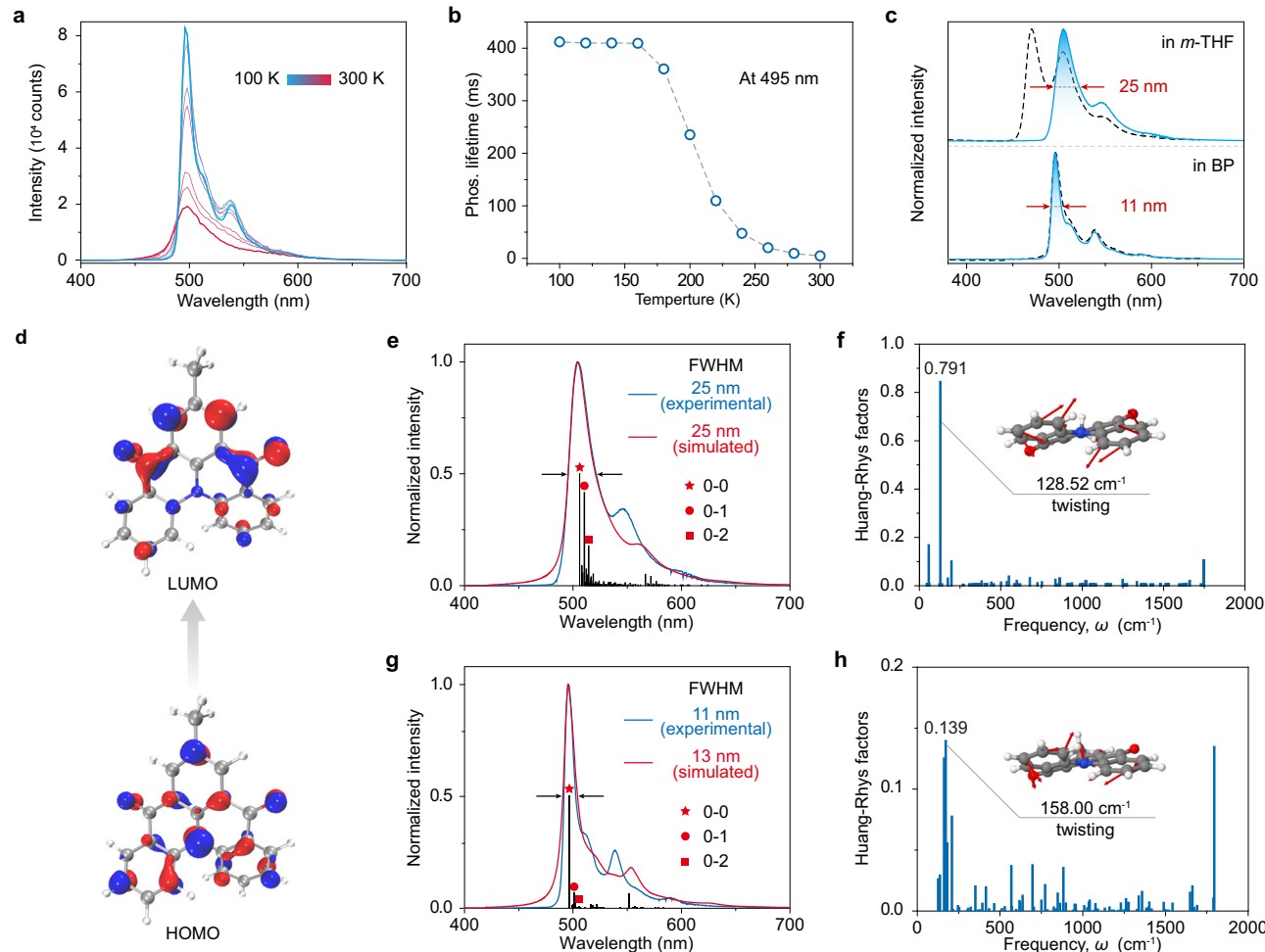

**Fig. 3 | Mechanism investigation for the narrowband phosphorescence.**
**a** Temperature-dependent steady-state PL spectra. **b** Temperature-dependent lifetimes of the emission at 495 nm. **c** Normalised steady-state PL (dotted lines) and phosphorescence (solid lines) spectra of 7-MQ in *m*-THF dilute solution ($1 \times 10^{-5}$ M, top) and BP matrix (bottom) at 77 K. Arrows indicate FWHM. **d** HOMO and LUMO orbits of the 7-MQ molecule with B3LYP/def2-SVP functional. **e** Experimental (blue) and simulated (red) phosphorescence spectra of the 7-MQ molecule in *m*-THF at 77 K. Arrows indicate FWHM. **f** Calculation of Huang−Rhys factors for the 7-MQ molecule in *m*-THF at 77 K. **g** Experimental (blue) and simulated (red) phosphorescence spectra of the 7-MQ molecule in BP matrix at 77 K. **h** Huang−Rhys factors for the 7-MQ molecule in BP matrix at 77 K. Source data are provided as a Source Data file.

MQ/BP phosphor is phosphorescence rather than TADF. From Fig. 3c, it was easily found that the phosphorescence in BP matrix was consistent with that in dilute *m*-THF solution at 77 K. From this we inferred that the phosphorescence in BP matrix stemmed from the isolated 7-MQ molecules. Subsequently, we investigated the energy transfer between the BP host and the 7-MQ guest. As shown in Supplementary Fig. 32, we found that both profiles of the PL spectrum of BP and absorption spectrum of 7-MQ overlapped well. Besides, the lowest triplet energy level is lower than that of the BP host matrix, enabling Dexter energy transfer from the host to guest molecules (Supplementary Fig. 33a). From the excitation spectrum monitoring the emission at 495 nm of the 7-MQ/BP phosphor (Supplementary Fig. 32), we speculated that there is efficient energy transfer between the host and guest molecules because of existence of a strong absorption band ranging from 300 to 400 nm of the BP host matrix, which is also confirmed by disappearance of the emission band (400–500 nm) from the BP host matrix when the 7-MQ molecule was doped in the BP host at low levels (Fig. 2a and Supplementary Fig. 33b).

As shown in Fig. 3c, the FWHM of the phosphorescence spectrum for 7-MQ in BP matrix is only 11 nm, which is much narrower than that in *m*-THF solution at 77 K. Conversely, dimethyl 2,2′-(*p*-tolylazanediyl)dibenzoate, the precursor of 7-MQ, is a flexible and open-loop molecule, which shows a broadband phosphorescence

profile with FWHM of 57 nm in dilute *m*-THF solution ($1 \times 10^{-5}$ M) at 77 K (Supplementary Fig. 34). These results indicated that a rigid and closed-loop molecular structure, as well as host−guest interaction, plays a vital role in narrowing the phosphorescence spectrum to enhance the colour purity. To discover the cause of the narrowband phosphorescence, we further conducted a series of theoretical calculations. As anticipated, the multiple resonance effect of the nitrogen atom and carbonyl group induces the localisation of HOMO and LUMO on different atoms (Fig. 3d), leading to non-bonding molecular orbitals. Consequently, the reorganisation energy of $T_1$ state is small (1349.3 cm$^{-1}$), endowing the molecule with reduced vibronic coupling and vibrational relaxation for narrowband phosphorescence generation. Indeed, the stimulated phosphorescence spectrum with FWHM of 25 nm can well match the experimental result (Fig. 3e). Taking the normal mode analysis together (Fig. 3f), we find that such a spectral progression is mainly ascribed to the twisting mode of molecular skeleton with a frequency of 128.52 cm$^{-1}$, which shows intense 0−1 and 0−2 vibrational peaks. After doped into BP matrix, these two peaks are largely reduced, further diminishing the FWHM to 13 nm, which is consistent with the experimental value. It is also proved by the changes in the Huang−Rhys factors, which characterise the intense of the vibrational peaks, e.g., decreasing from 0.791 to 0.139 (Fig. 3h) for the twisting mode of molecular

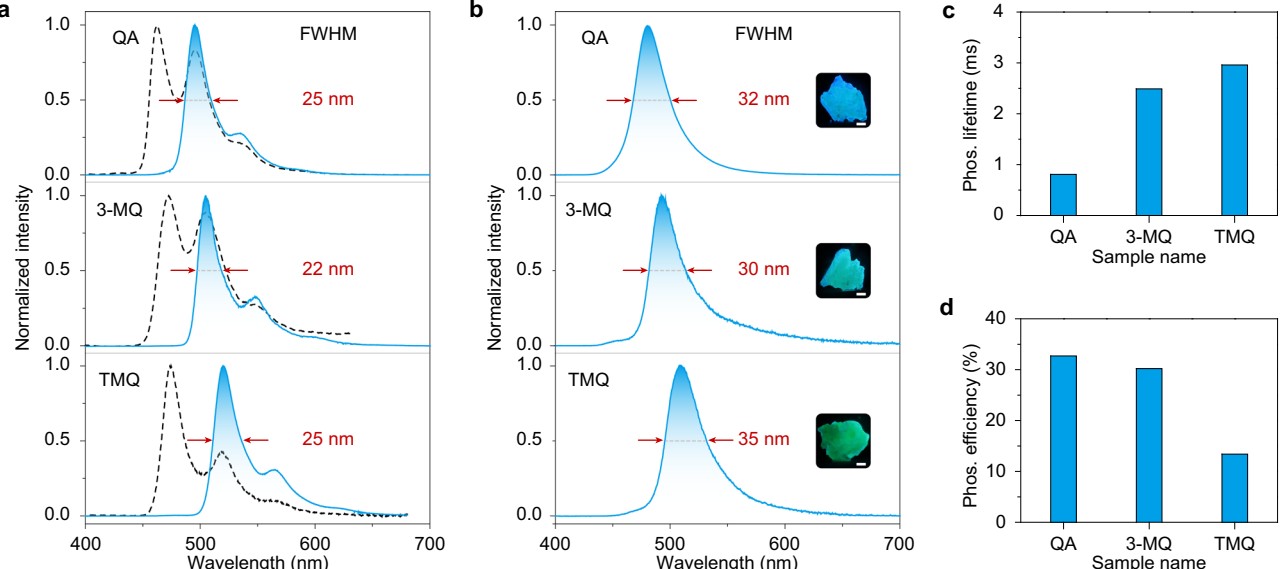

**Fig. 4 | Photophysical properties of the QA, 3-MQ, and TMQ molecules in *m*-THF and BP matrix. a** Normalised steady PL (dotted lines) and phosphorescence (solid lines) spectra of the QA, 3-MQ and TMQ molecules in dilute *m*-THF solution (1 × 10⁻⁵ M) at 77 K. **b** Phosphorescence spectra of the QA, 3-MQ and TMQ molecules in BP matrix under ambient condition. The ratios of the guest molecules in BP matrix is 1:1000. Arrows indicate FWHM. Insets are photographs taken under excitation by 380 nm UV light. The scale bar is 1 mm. **c, d** Phosphorescence lifetimes (**c**) and efficiency (**d**) of QA, 3-MQ, and TMQ in BP matrix under ambient conditions. Source data are provided as a Source Data file.

skeleton. Therefore, we concluded that the narrowband phosphorescence stemmed from the synergistic effect of the multiple resonance molecule structure and the suppression of twisting vibrations of the molecular skeleton by the BP host matrix.

## Extended experiments on design principle's universality

To prove the universality of our approach, a series of multiple resonance phosphorescence molecules, namely QA, 3-MQA and TMQ, were designed and synthesised. Firstly, we collected the steady-state PL and phosphorescence spectra of these molecules in dilute *m*-THF solution (1 × 10⁻⁵ M) at 77 K. As the number of the methyl groups increasing, the main bands of the PL spectra showed redshift from 460, 465 to 482 nm with nanosecond lifetimes (Supplementary Figs. 36 and 37 and Supplementary Table 6) for QA, 3-MQ and TMQ, respectively. As expected, narrowband phosphorescence of QA, 3-MQ and TMQ shows narrow FWHMs of 25, 22 and 25 nm in dilute solution at 77 K after a delayed time of 8 ms, respectively. The phosphorescence peaks at 494, 504 and 520 nm have corresponding lifetimes of 499.8, 443.1 and 713.0 ms (Supplementary Fig. 38 and Supplementary Table 7), respectively. Like 7-MQ, the molecules of QA, 3-MQ, and TMQ also showed narrowband phosphorescence with FWHMs of 32, 30 and 35 nm under ambient conditions after doped into BP matrix (Fig. 4b), of which corresponding phosphorescence lifetimes are 0.81, 2.49 and 2.96 ms, respectively (Fig. 4c). At 77 K, the FWHMs of the phosphorescent profiles for QA (11 nm), 3-MQ (14 nm) and TMQ (17 nm) become narrower (Supplementary Fig. 39), which are consistent with 7-MQ. With methyl modification, the phosphorescence colours of QA, 3-MQ and TMQ can be reasonably tuned from blue to green with emission peak changed from 480, 493, to 508 nm, respectively. The corresponding CIE coordinates for RTP are (0.115, 0.264), (0.201, 0.496) and (0.195, 0.644), respectively (Supplementary Fig. 40). The phosphorescence efficiencies of the QA, 3-MQ and TMQ doped into the BP matrix are 32.7%, 30.2% and 13.4% under ambient conditions, respectively (Fig. 4d). Like 7-MQ, the phosphorescence spectra of QA, 3-MQ, and TMQ in the BP matrix leave their profiles unchanged regardless of the variation of excitation wavelengths (Supplementary Fig. 41).

## Potential applications of the narrowband RTP

Organic phosphorescent materials exhibit a significant advantage in efficiently utilising excitons when stimulated by X-rays (Fig. 5a)[7]. Moreover, X-rays can be utilised to facilitate charge injection[42]. To further assess the potential applications of these phosphors, we initiated an investigation into the radioluminescence of 7-MQ. Under X-ray irradiation, we observed the emergence of narrowband delayed fluorescence, akin to profiles observed under UV-light excitation (Fig. 5b and Supplementary Fig. 42). Subsequently, we explored their potential application for scintillation in X-ray imaging, employing X-ray imaging equipment assembled by our team (Fig. 5c). Two uniform scintillators were prepared as films through a process involving melting and crystallising on a 3 × 3 cm² quartz sheet (Fig. 5d). As shown in Fig. 5e (left), we successfully obtained X-ray imaging of a pen. The X-ray image distinctly reveals the internal structure of the pen nib, including fine details such as the air hole. So, the use of narrowband emission not only enhances colour purity for display purposes but also minimises spectral overlap, thereby eliminating signal crosstalk between colour channels during imaging. Despite advancements, achieving chromatic X-ray imaging remains a considerable challenge. To address this, we developed a bilayer thin film incorporating 7-MQ and 4,9-dibromoisochromeno[6,5,4-*def*] isochromene-1,3,6,8-tetraone for chromatic X-ray imaging. Here, 7-MQ contributes to the blue channel, while 4,9-dibromoisochromeno[6,5,4-*def*] isochromene-1,3,6,8-tetraone governs the red channel. Upon X-ray irradiation, a chromatic X-ray image was successfully produced (Fig. 5e, right). It was found that regions with higher X-ray exposure exhibit a reddish-orange hue along the margins, while areas with lower X-ray dosage appear blue in the middle segment.

Concurrently, we also showcased the potential of the phosphors for a narrowband backlit display using a 365 nm UV lamp in ambient conditions (Supplementary Fig. 43). We assembled a backlit display device with a sandwich structure comprising 15 dots, each of which could be independently controlled to emit light or remain off via a single chip computer (Supplementary Fig. 43a). The surface of each LED was coated with the 7-MQ/BP phosphor. Through programmed control, numerical digits ranging from 0 to 9 were displayed

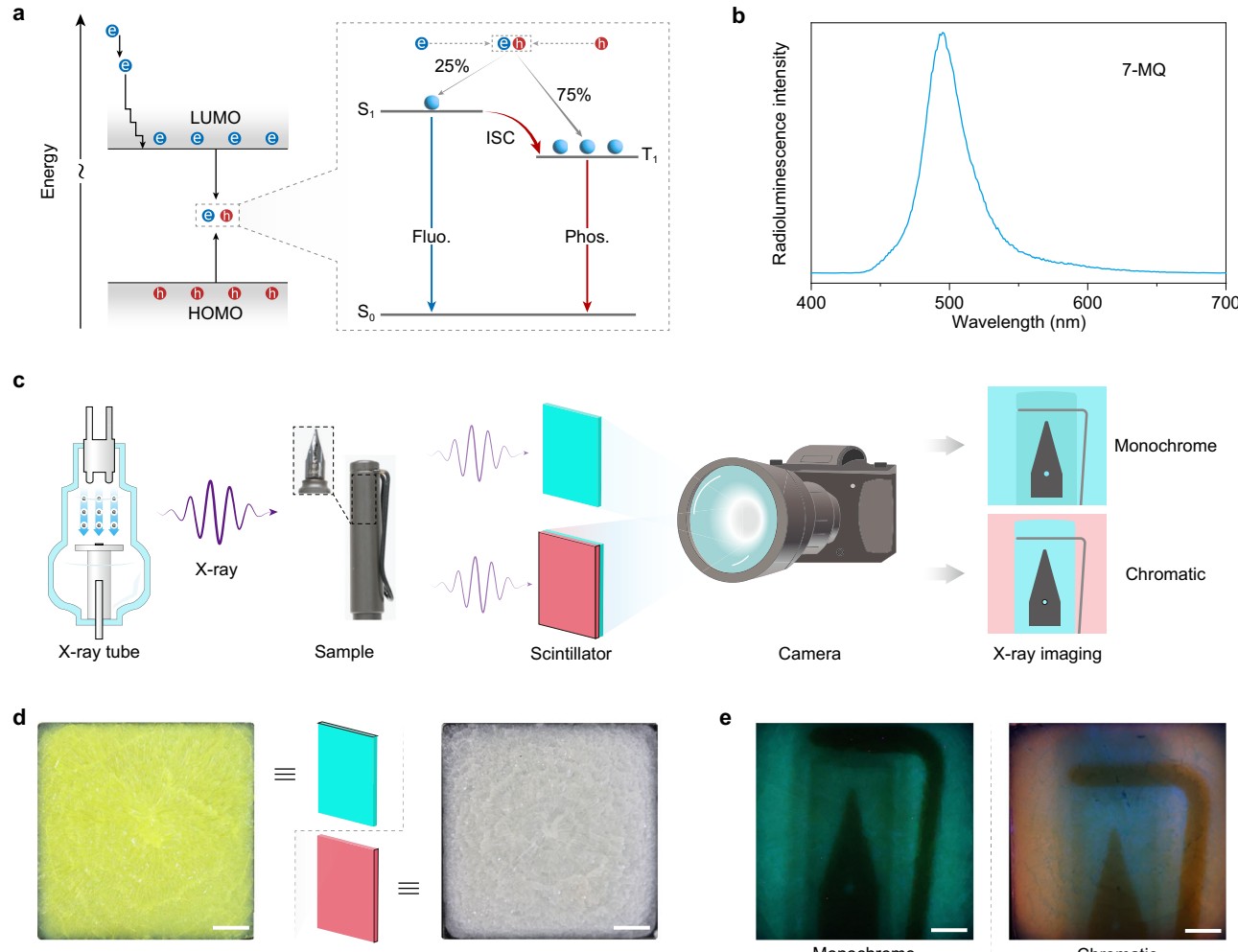

**Fig. 5 | Description of narrowband RTP materials for X-ray imaging. a** Proposed energy transfer processes for radioluminescence in organic molecules. The electrons and holes were generated after X-ray excitation. Excitons are produced in a ratio of 25% singlet excitons and 75% triplet excitons by the recombination of electrons and holes. **b** Normalised radioluminescence spectrum of the 7-MQ molecule in the BP matrix. **c** The schematic diagram of the equipment for the X-ray imaging. Monochrome and chromatic X-ray imaging of a pen were imaged with the single-layer and dual-layer scintillators. **d** Photography of X-ray scintillators (3 × 3 cm²) under sunlight. The yellow (left) and white (right) films are the 7-MQ and 4,9-dibromoisochromeno[6,5,4-*def*] isochromene−1,3,6,8-tetraone molecules in the BP matrix, respectively. **e** Monochrome and chromatic X-ray imaging of a pen. The images were taken with a Canon R5 camera (lens: EF 100 mm f/2.8 L IS USM). The scale bar is 5 mm. Source data are provided as a Source Data file.

(Supplementary Fig. 43b). This setup enables the presentation of various information with high colour purity.

## Discussion
In conclusion, we have reported a universal molecular design principle of multiple resonance effect to obtain narrowband emission for higher RTP colour purity. The phosphors exhibit narrowband phosphorescence with FWHMs of 30 nm after being doped into the BP matrix under ambient conditions, which show a lifetime of around 2.8 ms and an efficiency of 51.8%. At 77 K, the FWHMs are reduced to a lower level of 11 nm. With the modification of the methyl group, the colour of narrowband RTP can be controllably tuned from sky blue to green. Taking the theoretical results together, it was found that the restriction of twisting vibrations of the molecular skeleton by the rigid matrix plays a crucial role in narrowing phosphorescence bands. Additionally, narrowband RTP could be also obtained by irradiation with X-ray. Given this and narrowband feature, the potential applications for X-ray imaging and backlit display were demonstrated. This finding provides a

perspective to develop narrowband RTP materials, which will lay the foundation of the potential applications in the fields of sophisticated techniques of ultrahigh-definition display, high-resolution imaging, etc.

## Methods
### Materials
Unless otherwise noted, all reagents (purity > 99%) used in the experiments were purchased from commercial sources without further purification. Benzophenone was purified by column chromatography twice and recrystallisation.

### Molecular synthesis
The target molecules of 7-methylquinolino[3,2,1-*de*]acridine-5,9-dione (7-MQ), 3-methylquinolino[3,2,1-*de*]acridine-5,9-dione (3-MQ), quinolino[3,2,1-*de*]acridine-5,9-dione (QA) and 3,7,11-trimethylquinolino[3,2,1-*de*]acridine-5,9-dione (TMQ) were prepared in three steps from commercially available compounds and fully characterised by NMR and elemental analysis (Supplementary Figs. 3−18).

## Measurements

$^1$H and $^{13}$C nuclear magnetic resonance (NMR) spectra were measured with a JOEL NMR spectrometer (JNM-ECZ400S, 400 MHz, Japan). The chemical shift was relative to tetramethylsilane (TMS) as the internal standard. Elemental analysis was tested by a Vario EL Cube. Steady-state and delayed photoluminescence spectra were collected using Hitachi F-7100. The photoluminescence lifetimes and time-resolved emission spectra were collected on an Edinburgh FLS1000 fluorescence spectrophotometer equipped with a xenon arc lamp (Xe900), a nanosecond hydrogen flash-lamp (nF920), and a microsecond flash-lamp (µF900), respectively. Photoluminescence efficiency was obtained on a Hamamatsu Absolute PL Quantum Yield Spectrometer C11347. Powder X-ray diffraction (PXRD) patterns were collected on a Smartlab (3 kW) X-ray diffractometer of a Japanese brand. The radioluminescence (RL) spectra were collected on an Edinburgh FS5 fluorescence spectrophotometer with an X-ray tube (Tungsten target, Moxtex). Photographs were taken with a Cannon EOS 90D camera. Unless other noted, all photophysical properties of the solids were collected at 298 K with relative humidity (RH) of ≈30% in air.

## Preparation of the host–guest doped phosphors

Firstly, the dichloromethane solutions of the emitters with concentrations of 1 mg mL$^{-1}$ were prepared. The emitter's solutions with different volumes were added into the quartz tubes, in which the dichloromethanes were evaporated by heating. Then, benzophenone (100 mg) was added to the quartz tube. After that, the mixtures were heated to liquation. After cooling the liquids to room temperature, the phosphors were obtained.

## Computational details

The QM/MM models for doping were constructed by cutting a 5 × 5 × 5 supercell from experimental crystal structures. Two molecules at the center of the supercell were replaced by one 7-MQ molecule. The 7-MQ molecule was defined as the active QM part, and the surrounding area was set as rigid, constituting the MM part. The QM/MM calculations were performed using the ChemShell 3.7[43] package, which interfaced with ORCA 5.0.2[44] for QM and DL_POLY[45] with the general Amber force field (GAFF)[46] for the MM part. The atomic partial charges were generated by the restrained electrostatic potential (RESP)[47] method. An electrostatic embedding scheme was applied in QM/MM calculations. The QM calculations of the equilibrium configuration and harmonic vibrational frequency were performed at (TD)B3LYP/def2-SVP level. The molecule in solvent environment calculations were performed by ORCA 5.0.2 at (TD)B3LYP/def2-SVP level, and the solvation model based on the quantum mechanical charge density SMD[48] (THF) was used to include solvent effects. The Huang–Rhys factors and simulated spectra were obtained using the MOMAP program[49].

## Reporting summary

Further information on research design is available in the Nature Portfolio Reporting Summary linked to this article.

## Data availability

The data that support the findings of this study are available from the corresponding author upon request. Source data are provided with this paper.

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

## Acknowledgements

This work is supported by the National Key R&D Program of China (2020YFA0709900 (W.H.)) and the National Natural Science Foundation of China (62288102 (W.H.), 22371123 (H.S.), 22275085 (H.M.), 62305276 (X.W.)).

## Author contributions

X.Y., Y.L. and Z.A. conceived the experiments. X.Y., H.S. and Z.A. prepared the paper. X.Y., Y.L., B.W., Z.Z., C.Z., X.Z. and M.T. were primarily responsible for the experiments and measurements. Z.Y. and H.M. contributed to TD-DFT calculations. X.W., H.S., H.M, Z.M. and W.H. gave suggestions for the manuscript. All authors contributed to the data analyses.

## Competing interests

The authors declare no competing interests.
