## [Peer Review File · Nature Communications]

Narrowband room temperature phosphorescence of closed-loop molecules through the multiple resonance effectReviewer #1 (Remarks to the Author):

This paper describes the narrowband room-temperature phosphorescence by MR effect. The pure organic phosphors with RTP was obtained. However, the color purity of RTP showed in the benzophenone matrix below 1% doping. The author did not shown any application for RTP. In my opinion, the long lived organic phosphor have limited application in the organic electronics and do not interested to related reader. Therefore, I do not recommend this paper to publish in Nature Communication.

Reviewer #2 (Remarks to the Author):

The manuscript reports the synthesis of a new series of organic molecules to obtain narrowband RTP, which is of scientific and technological interest for optoelectronic applications in general. In Supplementary Table 1 the authors show a large list of papers with FWHMs values of RTP materials to highlight the values achieved in this manuscript. My main concern is related to conclusion of the phosphorescence nature of the molecules emission, which in my opinion is not completely supported by the experimental results and discussion. Theoretical calculations could also be better explored in this case.

The reduction of the steady-state PL intensity by increasing the temperature (Fig 3a) is expected for organic emitting molecules. The continuous reduction of the lifetime by increasing temperature (Fig 3b) is also observed for TADF materials. Therefore, these results do not allow to undoubtedly conclude: "According to the above results, we concluded that the emission nature of the 7-MQ/BP phosphor is phosphorescence rather than TADF." as stated by the authors. Obviously that lifetimes ranging from 405 to 2.8 ms are not common for TADF emission, but there must be additional evidences or explanations to support the nature of phosphorescence emission.

The steady PL and delayed spectra overlap observed in Fig 2a (RT) and Fig 3c (77 K) is usually the case of TADF emission, where S1 and T1 energy levels need to be closely aligned.

The favored phosphorescent decay from T1 can be supported from the calculations. Nevertheless, the overlap with the experimental spectra presented in Figs 3e, 3g and on the Supplementary Fig. 28. is impressive.

Experimentally, the authors could measure the delayed spectra (at a delay time of 8 ms) of the 7-MQ/BP as a function of temperature, from 300 K down to 80 K. The integrated intensity of the spectra should decrease in case of TADF and increase in case of phosphorescence.

The authors used a Hamamatsu Absolute PL Quantum Yield Spectrometer to measure the phosphorescence PLQY. How do they avoid the steady PL emission? Does the system allow to apply a delay time? They should provide more details.

Reviewer #3 (Remarks to the Author):

This manuscript by Z. An and coworkers reports the fabrication of narrowband organic room-temperature phosphorescence by multiple resonance effect, which has small FWHM of 30 nm. In the area of organic room-temperature phosphorescence and afterglow materials, narrowband emission has been rarely reported, as summarized by the authors in supporting information. By changing the substituent group of the luminescent dopants, the emission colors can be regulated from blue to green. The authors have also shown some interesting demonstration of the function of the obtained narrowband materials.

There are some additional comments. (1) Benzophenone has been used as matrix for material fabrication. To reveal the role of matrix, the study of the photophysical properties of the materials using different matrices may be helpful. (2) Why does the spectrum broaden significantly with the increasing doping ratio in Fig. 2a? (3) How about the stability of narrowband RTP at different atmospheres? (4) Metals or heavy atoms have not been involved in the materials, whereas the emission lifetimes are in millisecond scale. Is this caused by the structures of luminescent dopants, or benzophenone matrices, or some other reasons?

The contents of this study is well organized and this manuscript is easy to follow. I support the acceptance of this manuscript after rational revisions.

Point-by-Point Response to Referees

Reviewer #1:

This paper describes the narrowband room-temperature phosphorescence by MR effect. The pure organic phosphors with RTP was obtained. However, the color purity of RTP showed in the benzophenone matrix below 1% doping. The author did not shown any application for RTP. In my opinion, the long lived organic phosphor have limited application in the organic electronics and do not interested to related reader. Therefore, I do not recommend this paper to publish in Nature Communication.

Response: We truly understand the concern of the reviewer. Since the end of the last century, organic phosphorescent materials have received considerable attention in organic electronics owing to their unique properties (*Chem. Rev.* 2002, 102, 2357; *Chem. Soc. Rev.* 2011, 40, 2508.). For instance, to eliminate the background fluorescence, the phosphorescence materials were applied to bioimaging, sensing or information anticounterfeiting for their long-lived emission lifetimes in the scale of region from μs to s (*Nat. Mater.* 2009, 8, 747; *Nat. Chem.* 2023, 15, 83; *Nat. Mater.* 2015, 14, 685.). With the potential of high exciton utilization ($\sim 100\%$), they have been used as emitters to enhance the device performance in organic light-emitting diodes (OLEDs) (*Nature* 1998, 395, 151; *Science* 2017, 356, 159; *Chem. Rev.* 2021, 121, 7249; *Nat. Commun.*, 2023, 14, 1678.). Recently, more effort was devoted to developing organic luminescent materials with narrowband emission in specially, which was beneficial to improving color purity of OLEDs and imaging resolution. So far, some narrowband organic emitters, including conventional fluorescence, phosphorescence based on metal complex and thermally activated delayed fluorescence, etc., have been developed. Very recently, purely organic room temperature phosphorescence (RTP) materials as a new type of luminescent materials are emerging. However, there is rare report on RTP molecules with narrowband emission features. In this manuscript, we focus on the strategy for designing the RTP materials with narrowband emission. As expected, we obtained narrowband RTP in a series of quinolino [3,2,1-*de*]acridine-5,9-dione by multiple resonance effect.

As suggested by the reviewer, we tried our best to add some potential applications of the narrowband RTP materials in the past months. It is worth noting that organic phosphorescent materials have the advantage of high exciton utilization under excitation by high-energy rays (Supporting Figure 1a, *Nat. Photon.* 2021, 15, 187.). Therefore, we first investigated the potential application of the model phosphorescent material (7-MQ) for X-ray imaging. From Supporting Figure 1b, it was found that the radioluminescence spectrum with the narrowband feature is identical to the photoluminescent spectrum by UV-light excitation, demonstrating the potential application of 7-MQ in the radiation field. Firstly, we prepared a uniform film with 7-MQ in BP matrix as an X-ray scintillator. With the equipment set up by ourselves (Supporting Figure 1c), we obtained the X-ray imaging of a pen. The internal profile of the pen nib can be clearly distinguished from the image, even the air hole in the nib. Notably, narrowband emission is not only beneficial to improve color purity for display, but also avoids overlap of the emission spectra to eliminate signal crosstalk from the color channels for imaging. So far, it is still a challenge to obtain a chromatic X-ray image. Herein, we prepared a bilayer thin film with 7-MQ and 4,9-Dibromoisochromeno[6,5,4-*def*]isochromene-1,3,6,8-tetraone for chromatic imaging by X-ray irradiation. The 7-MQ molecule is attributed to the blue channels, while 4,9-

dibromoisochromeno[6,5,4-def] isochromene-1,3,6,8-tetraone is responsible for the red channel. From Supporting Figure 1d, a chromatic X-ray image was obtained. It is easily found that red-orange appears at the margin because there is a lot of X-ray radiation. In the middle part, it is blue due to a low dose of X-ray.

Supporting Figure 1. Potential applications for X-ray imaging of narrowband RTP materials. **a**, Radioluminescence mechanism of organic compound by X-rays. **b**, Radioluminescence spectrum of the 7-MQ molecule in the BP matrix. **c**, Schematic diagram of the equipment of the X-ray imaging with 7-MQ/BP and the X-ray imaging. **d**, Demonstration of chromatic X-ray imaging. Note. The red and blue-green films are the 4,9-dibromoisochromeno[6,5,4-def] isochromene-1,3,6,8-tetraone and 7-MQ molecules doped into the BP matrix, respectively. The images were taken with a Cannon R5 camera.

Simultaneously, we also demonstrated the narrowband backlit display using a 365 nm UV lamp under ambient environment (Supporting Figure 2). A sandwich structure backlit display device with 15 dots was constructed, of which light dots can be turned on and off by a single chip computer (Supporting Figure 2a). On the surface of each LEDs, the 7-MQ/BP phosphor was sprinkled. With the program control, the numbers from 0 to 9 were shown (Supporting Figure 2b). Various information can be displayed with high color purity.

Supporting Figure 2. Schematic of the backlit display of the 7-MQ/BP materials.

In summary, metal-free organic room-temperature phosphorescence (RTP) materials as a new type of luminogens have made rapid development in improving RTP performance through molecular engineering recently. However, researchers have yet to report a purely RTP molecule with full width at half maxima (FWHM) of less than 50 nm in the past decades. Our work mainly aims to propose a strategy to obtain narrowband RTP. The phosphors show narrowband phosphorescence with FWHMs of 30 nm under ambient conditions, of which the RTP efficiency reaches 51.8%. As suggested by the reviewer, we demonstrated the potential applications of the RTP phosphors for chromatic X-ray imaging and backlit display.

We hope our clarification is acceptable to the reviewer.

Reviewer #2:

The manuscript reports the synthesis of a new series of organic molecules to obtain narrowband RTP, which is of scientific and technological interest for optoelectronic applications in general. In Supplementary Table I the authors show a large list of papers with FWHMs values of RTP materials to highlight the values achieved in this manuscript.

My main concern is related to conclusion of the phosphorescence nature of the molecules emission, which in my opinion is not completely supported by the experimental results and discussion. Theoretical calculations could also be better explored in this case.

Response: We appreciate the positive comments from the reviewer.

Comments #1: The reduction of the steady-state PL intensity by increasing the temperature (Fig 3a) is expected for organic emitting molecules. The continuous reduction of the lifetime by increasing temperature (Fig 3b) is also observed for TADF materials. Therefore, these results do not allow to undoubtedly conclude: "According to the above results, we concluded that the emission nature of the 7-MQ/BP phosphor is phosphorescence rather than TADF" as stated by the authors. Obviously that lifetimes ranging from 405 to 2.8 ms are not common for TADF emission, but there must be additional evidences or explanations to support the nature of phosphorescence emission.

Response: Thanks for the professional review from the reviewer. We truly understand the concern of the reviewer in the phosphorescence nature of our materials. Herein, we conducted a series of experiments to look into the emission properties of our materials in the following aspects:

- (1) Temperature-dependent photophysical properties
- (2) Analysis between the energy levels of host and guest molecules
- (3) Investigation of TADF emission for the 7-MQ molecule in matrices

(1) Temperature-dependent photophysical properties

Thermally activated delayed fluorescence (TADF):

TADF is a unique type of fluorescence emission, which usually contains two-part lifetime emission on the nanosecond and microsecond scale at room temperature. The long-lived part of fluorescence (DF) is caused by the reverse intersystem crossing (RISC) of the triplet exciton to the excited singlet state, whereas the radiative transition of the singlet exciton without intersystem crossing process is responsible for the nanosecond part of the fluorescence. The RISC rate is proportional to the Boltzmann factor, $\exp(-\Delta E_{ST} / k_B T)$, which weights the relative filling of T_1 and S_1 at thermal equilibrium (*Organic Electronics: Foundations to Applications*, Stephen R. Forrest, 2020 by Oxford University Press; *Nat. Rev. Mater.* 2018, 3, 18020.).

$$k_{RISC} \propto \exp(-\Delta E_{ST} / k_B T)$$

Where ΔE_{ST} is the singlet-triplet exchange energy; k_B is the Boltzmann constant; T is the temperature. Notable, the RISC process relates to the temperature, which enhanced the triplet exciton transfer to excited singlet state by thermal stimulation. Because of Hund's rule, which states that the T_1 energy level is always lower than the S_1 energy level, the RISC process is notably endothermic. As a result, molecules do not exhibit DF at low temperatures, and are gradually enhanced with temperature rises. So TADF molecules exhibit a unique change tendency in emission intensity and lifetime with temperature due to their unique exciton transfer process. **Typically, thermal stimulation generally increases the lifetime of DF**

as the temperature rises. However, this lifetime is subsequently shortened due to the enhancement of non-radiative transitions of the triplet exciton (Supporting Figure 3, *Nature* 2012, 492, 234; *Nat. Photon.* 2018, 12, 98; *Adv. Mater.* 2020, 32, 2003911; *Nat. Commun.* 2023, 14, 419; *Nat. Commun.* 2023, 14, 2394. *Nat. Commun.* 2023, 14, 2564.). For most TADF molecules, there is the longest DF emission lifetime in the temperature ranging from 200 to 300 K.

Supporting Figure 3. a, Schematic of the exciton conversion in TADF molecules. Note: The Exc., PF, DF, ISC and RISC represent excitation, promote fluorescence, delay fluorescence, intersystem crossing and reverse intersystem crossing, respectively. **b**, Temperature-dependent lifetime delay curves of TADF molecule (4CzIPN) in the *m*CP matrix in a previous paper (*Nature* 2012, 492, 234.). Supporting Figure 3b reproduced with permission from *Nature* 2012, 492, 234. (Copyright 2012 Springer Nature).

Phosphorescence

For phosphorescence materials, the triplet exciton goes back to the ground state by radiative (phosphorescence) and non-radiative transitions process. The radiative transition rates can be expressed as (*Chem. Rev.* 1966, 66, 199; *J. Phys. Chem.A.* 2007, 111, 10490.):

$$k_p \approx \bar{\nu}^2 f = \bar{\nu}^2 \frac{8\pi m_e \bar{\nu}}{3\hbar e^2} |\bar{\mu}_{T_1 \rightarrow S_0}|^2$$

$$\bar{\mu}_{T_1 \rightarrow S_0} = \sum_n \lambda_n \times \mu_{S_n \rightarrow S_0} + \sum_m \lambda_m \times \mu_{T_m \rightarrow T_1}$$

$$\lambda_n = \frac{\langle S_n | \hat{H}_{soc} | T_1 \rangle}{E_{S_n - T_1}}$$

$$\lambda_m = \frac{\langle T_m | \hat{H}_{soc} | S_0 \rangle}{E_{T_m - S_0}}$$

$$H_{soc} = -\frac{Z^4 e^2}{8\pi \epsilon_0 m_e^2 c^2} LS$$

where $\bar{\nu}$ is the wavenumber corresponding to the phosphorescence energy, f is the oscillator strength of phosphorescence from T_1 , $\bar{\mu}_{T_1 \rightarrow S_0}$ is the transition dipole moment between the higher lying singlet excited state (S_n) and S_0 , $\langle S_n | \hat{H}_{soc} | T_1 \rangle$ is the SOC between S_n and T_1 , $E_{S_n - T_1}$ is the energy difference between S_n and T_1 , $\mu_{T_m \rightarrow T_1}$ is the transition dipole

moment between the higher lying triplet excited state (T_m) and T_1 , $\langle T_m | \hat{H}_{soc} | S_0 \rangle$ is the SOC between S_0 and T_m , and $E_{T_m-S_0}$ is the energy difference between S_0 and T_m . ϵ_0 is vacuum dielectric constant, m_e is the mass of the electron, c is the speed of light and Z is the atomic number. The vector part is the dot product of the electron's angular momentum L and its spin S . It can be found that there is correlation between temperature and the radiative transition process.

The non-radiative transition of phosphorescence at 77 K can be represented as an equation (Chem. Phys. Lett. 1973, 22, 186; App. Phys. Rev. 2022, 9, 011304.):

$$k_{nr}(77 \text{ K}) \approx \frac{2\pi}{\hbar} \sum_p |\partial \langle T_1 | \hat{H}_{soc} | S_0 \rangle \partial Q_p|^2 \exp(\tilde{\tau}\omega_p)/2\omega_p FC$$

where the Q_p is the coordinate of the chromophore;

$|\partial \langle T_1 | \hat{H}_{soc} | S_0 \rangle \partial Q_p|^2$ indicates the magnitude of the change in SOC between T_1 and S_0 depending on the change in coordinate due to the p^{th} vibrations;

$\exp(\tilde{\tau}\omega_p)/2\omega_p$ is a vibrational factor describing how the amplitude of vibrations with lower frequency becomes large under the same temperature;

FC is a function involving the nuclear vibration overlap integral and the energy gap law.

On the basis of this definition, $k_{nr}(RT)$ is expressed in terms of the vibrational factor at T $P(T)$ as (J. Phys. Chem. A 2021, 125, 885.):

$$k_{nr}(T) \propto \frac{2\pi}{\hbar} \sum_p |\partial \langle T_1 | \hat{H}_{soc} | S_0 \rangle \partial Q_p|^2 P(T) FC$$

The phosphorescence lifetime and quantum efficiency can express:

$$\tau_p = \frac{1}{k_p + k_{nr}}$$

$$\Phi_p = \Phi_{isc} \frac{k_p}{k_p + k_{nr}}$$

The equations show that there is a greater correlation between temperature and the characteristics of phosphorescence. **Due to the enhancement of the non-radiative transition for the triplet exciton with increasing temperature, phosphorescence materials exhibit a downward tendency for their temperature-dependent emission intensity and lifetime.**

As shown in Figures 3a and 3b in the revised manuscript, the phosphorescence lifetime and intensity of 7-MQ/BP spiral downward with increasing temperature. **This temperature-dependent luminescence intensity and lifetime tendencies align with the phosphorescence feature.** Additionally, as the temperature increased, the temperature-dependent delayed spectra also showed a downward tendency (please see Comment #4). In conclusion, the photophysical characteristics of the 7-MQ/BP phosphor match with the nature of the phosphorescence materials.

(2) Analysis between the energy levels of host and guest molecules

It is noteworthy that the photoluminescent nature of some organic molecules depends on the environment, especially for TADF molecules. For example, the DABNA-1 molecule shows TADF in the mCP host matrix. However, there is no TADF when the guest molecule of DABNA-1 is doped into the DPEPO matrix (Supporting Figure 4, Figure 3 in *Nat. Photon.* 2021, 15, 780.), which was ascribed to the lack of an exciplex-like interaction between the host and guest molecules. This exciplex is a middle state for promoting reversible intersystem crossing from the excited triplet to the singlet states, which can boost TADF emission.

Supporting Figure 4. a, Energy alignment of DPEPO and DABNA-1 molecules. b, Fluorescence decay curves for DEEPO in DABNA-1 (*Nat. Photon.* 2021, 15, 780.). Supporting Figure 4b reproduced with permission from *Nat. Photon.* 2021, 15, 780. (Copyright 2021 Springer Nature).

Supporting Figure 5. Energy alignment BP and 7-MQ molecules. Orbital energy alignment (a) and schematic of the state energy alignment (b) for BP and 7-MQ.

To rule out the TADF feature of emission, herein we further investigate the host-guest interaction. First, we measured the HOMO and LUMO levels for BP and 7-MQ molecules by cyclic voltammetry, respectively (see Supplementary Figure 26a and 26b). According to the

method from literature (ref: *Nat. Photonics* 2021, 15, 780; *Org. Electron.* 2011, 12, 2047.), the energy level of S_1/T_1 and S_2/T_2 for the host and guest emitter were obtained, which are 2.85 and 3.26 eV, respectively (Supporting Figure 5). From Supporting Figure 5, the $S_1(\text{CT})$ and $S_2(\text{CT})$ are higher than the $S_1(\text{LE})$ of 7-MQ by 0.21 and 0.62 eV, indicating that the 7-MQ/BP is a strictly thermal forbidden process. Thus, we conclude that the 7-MQ molecule fails to generate TADF in the BP matrix.

(3) Investigation of TADF emission for the 7-MQ molecule in matrices

To demonstrate the difference between phosphorescence and TADF of the 7-MQ molecule, *mCP* and DPEPO host molecules were chosen. The energy levels of *mCP* and DPEPO were referred from previous work (*Nat. Photon.* 2021, 15, 780.). There exist thermally accessible processes for the 7-MQ in *mCP* and DPEPO matrices due to the small energy gap between the $S_1(\text{CT})$ in *mCP* (0.06 eV) and DPEPO (0.04 eV) and $S_1(\text{LE})$ of the 7-MQ molecule, respectively (Supporting Figure 6a and 7a). From Supporting Figures 6b and 7b, it is found that the emission decay curves exhibit double-exponential for the 7-MQ in *mCP* and DPEPO matrices under ambient conditions, indicating there are TADF components for emission. There is an obvious difference in the steady state spectra of the 7-MQ in *mCP* and DPEPO matrices at 298 and 77 K (Supporting Figure 6c and 7c). Besides the fluorescent emission with peaks at 484 and 490 nm, new emission band appears for the 7-MQ in *mCP* and DPEPO after cooled to 77 K. The emission lifetimes are up to 530.6 and 555.4 ms for the 7-MQ in *mCP* and DPEPO (Supporting Figure 6d and 7d), respectively, which were ascribed to phosphorescence. When the 7-MQ was doped into BP matrix, there existed different emission properties. As shown in Figure 3c in the revised manuscript, the steady-state and delay spectra of the 7-MQ in BP are overlap. There is only one emission band with a peak at 495 nm. As temperature decreased from room temperature to 77 K, the emission intensity increases but no change for emission band (Figure 3a in the revised manuscript). Therefore, we reasoned that the emission band at around 495 nm corresponds to phosphorescence in BP matrix.

Supporting Figure 6. **a**, Schematic of the energy alignment for 7-MQ in *mCP*. **b**, Normalized steady-state and delay photoluminescence spectra (with a delay gate of 8.0 ms) of 7-MQ/*mCP* at 298 and 77 K. **c**, **d**, Lifetime time decay curves of 7-MQ/*mCP* at 298 and 77 K.

Supporting Figure 7. **a**, Schematic of the energy alignment for 7-MQ in DPEPO. **b**, Normalized steady-state and delay photoluminescence spectra (with a delay gate of 8.0 ms) of 7-MQ/DPEPO at 298 and 77 K. **c**, **d**, Lifetime time decay curves of 7-MQ/DPEPO at 298 and 77 K.

Supporting Figure 8. **a**, Temperature-dependent steady-state photoluminescence spectra of the 7-MQ in the *mCP* matrix. **b**, Fluorescence (484 nm) and phosphorescence (512 nm) intensity as a function of temperature.

Subsequently, we measured the temperature-dependent steady-state photoluminescence spectra of the 7-MQ doped in *mCP* solid. As shown in Supporting Figure 8, there is a significant difference of emission intensity for the fluorescence (484 nm) and phosphorescence (512 nm) peaks. The intensity of the fluorescence peak shows an upward tendency from 140 to 220 K, while phosphorescence intensity continues to decrease in the range of 80-300 K. The upward tendency of fluorescence is attributed to the thermally activated exciton transition from the triplet to singlet states. The temperature-dependent phosphorescence intensity of the 7-MQ in the *mCP* shows a similar tendency with the 7-MQ in the BP matrix.

In summary, from the above experimental results and analysis, we believe that the emission of 7-MQ at 495 nm is phosphorescence in the BP matrix. We hope that our clarification is acceptable for the reviewer.

Comments #2: The steady PL and delayed spectra overlap observed in Fig 2a (RT) and Fig 3c (77 K) is usually the case of TADF emission, where S_1 and T_1 energy levels need to be closely aligned.

Response: Thanks for the comment by the reviewer. From Figure 3c in the revised manuscript, the 7-MQ molecule has obvious fluorescence (2.64 eV) and phosphorescence (2.46 eV) bands in the *m*-THF solution at 77 K, of which ΔE_{st} is 0.18 eV. Meanwhile, the phosphorescence and fluorescence of the 7-MQ molecules can be distinguished in the *mCP* and DPEPO matrices at 77 K (Supporting Figure 6 and Figure 7), and the ΔE_{st} is 0.14 eV both in *mCP* and DPEPO matrices. However, there is an overlap between the steady-state and delay spectra for the 7-MQ molecule in the BP matrix at room temperature and 77 K, indicating that the triplet excitons of the 7-MQ molecule can be efficiently generated for phosphorescence in the BP matrix. Herein, we reasoned that efficient triplet-triplet energy transfer between the host and guest molecules and enhanced spin-orbit coupling of the 7-MQ molecule in the BP matrix play critical roles in boosting phosphorescence. Notable, BP matrix is a typical molecule with pure phosphorescence in solid state (Supporting Figure 9 and Supplementary Figure S24), which is beneficial to enhance phosphorescence emission with Dexter energy transfer.

Supporting Figure 9 a, Normalized steady-state and delay spectra of the BP molecule in m-THF solution at 77 K. **b**, Proposed energy transfer processes for 7-MQ/BP. Note that Ex., ISC, ET and Phos. are the abbreviation of excitation, intersystem crossing, energy transfer and phosphorescence, respectively.

Comments #3: The favored phosphorescent decay from T_1 can be supported from the calculations. Nevertheless, the overlap with the experimental spectra presented in Figs 3e, 3g and on the Supplementary Fig. 28. is impressive.

Response: We sincerely appreciate the comments by the reviewer. Indeed, an overlap between the calculation and experiment results further confirms the phosphorescence feature of the emission. Taking the calculation results together (Figs 3f and 3h), it also gives a deep insight into the reason for narrowband phosphorescence emission. The suppression of twisting vibrations of molecular skeleton by BP host matrix plays a critical role in generating narrowband phosphorescence.

Comments #4: Experimentally, the authors could measure the delayed spectra (at a delay time of 8 ms) of the 7-MQ/BP as a function of temperature, from 300 K down to 80 K. The integrated intensity of the spectra should decrease in case of TADF and increase in case of phosphorescence.

Response: We appreciate the valuable comments by the reviewer. As suggested, we collected the delayed spectra of the 7-MQ in the BP from 80 to 300 K. From Supporting Figure 10, it can be found that the emission intensity of the 7-MQ in the BP exhibited a gradually increased tendency from 300 to 80 K, indicating the phosphorescence nature of emission from the 7-MQ/BP solid.

Supporting Figure 10 Temperature-dependent delayed photoluminescence spectra (a) and emission intensity (b) of the 7-MQ in the BP matrix.

Comments #5: The authors used a Hamamatsu Absolute PL Quantum Yield Spectrometer to measure the phosphorescence PLQY. How do they avoid the steady PL emission? Does the system allow to apply a delay time? They should provide more details.

Response: We truly understand the concern of the reviewer. For Hamamatsu Absolute PL Quantum Yield Spectrometer, there is no system to manipulate a delay time. In this manuscript, there is only phosphorescence emission for all samples under ambient conditions, which is shown in Fig. 2a in the revised manuscript. So, the phosphorescent quantum yield is equal to the steady-state PL quantum yield in this manuscript.

.

Reviewer #3:

This manuscript by Z.An and coworkers reports the fabrication of narrowband organic room-temperature phosphorescence by multiple resonance effect, which has small FWHM of 30 nm. In the area of organic room-temperature phosphorescence and afterglow materials, narrowband emission has been rarely reported, as summarized by the authors in supporting information. By changing the substituent group of the luminescent dopants, the emission colors can be regulated from blue to green. The authors have also shown some interesting demonstration of the function of the obtained narrowband materials. There are some additional comments.

Response: We appreciate the positive comments from the reviewer.

Comments #1: Benzophenone has been used as matrix for material fabrication. To reveal the role of matrix, the study of the photophysical properties of the materials using different matrices may be helpful.

Response: Thanks for the professional comment of the reviewer. Indeed, the emission nature of the 7-MQ molecule is highly dependent on the host matrix properties such as HOMO, LUMO and triplet state energy levels. Here, the 7-MQ molecule was doped into *m*CP and DPEPO by melting crystallization. As shown in Supporting Figure 11a, these doped compounds demonstrated narrowband emission under ambient conditions. 7-MQ has two segments of emission lifetimes in both *m*CP and DPEPO matrices. One is on the nanosecond scale, while the other is on the microsecond scale, which is consistent with TADF emission characteristics (Supporting Figure 11b). To investigate the difference of the 7-MQ molecule in the *m*CP, DPEPO and BP, we compared energy level of the emission between host and guest. The emission properties of MR-TADF molecules with larger ΔE_{ST} are associated with the energy level of host and guest. The TADF emission highly relies on a host-emitter interaction (small energy difference between $LE(S_1)$ of the emitter and $CT(S_1/T_1)$ for host and guest) for promoting reversible intersystem crossing (RISC) of the emitter. From Supporting Figure 11c, it can be found that there is an energy gap of 0.21 eV between $CT(S_1/T_1)$ and $LE(S_1)$ in the 7-MQ/BP, thus the RISC is a thermally forbidden process. However, there is a tiny energy gap between the $LE(S_1)$ of the emitter and $CT(S_1/T_1)$ for the host and guest in *m*CP, DPEPO matrices, which promotes the RISC process for the emitter. So there is only TADF observed in *m*CP and DPEPO matrices. Therefore, we conclude that the emission nature of the emitter highly depends on the energy level of the matrices.

Supporting Figure 11. **a**, Normalized steady-state photoluminescence spectra of the 7-MQ in the *m*CP (blue line) and DPEPO (red line) under ambient conditions. **b**, Lifetime decay curves for the 7-MQ in the *m*CP (blue line) and DPEPO (red line) under ambient conditions. Schematic of the state energy alignment for 7-MQ in the BP (**c**) and *m*CP as well as DPEPO (**d**).

Comments #2: Why does the spectrum broaden significantly with the increasing doping ratio in Fig. 2a?

Response: From Figure 2a, the gradual broad emission profile with a doped ratio increase is owing to the rise of emission around 550-650 nm. Notably, the BP matrix does not exhibit the emission band in this range. Therefore, we reasoned that the 550-650 nm emission band might originate from the guest molecule. Then we measured the emission spectrum of the 7-MQ matrix in solid state. From Supporting Figure 12, it can be found that the pure guest spectrum matches with the raised region, indicating that the broad spectrum is owing to aggregation of the guest molecule.

Supporting Figure 12. Normalized steady-state photoluminescence spectrum of the 7-MQ

powder.

Comments #3: How about the stability of narrowband RTP at different atmospheres?

Response: Thanks for the comments of the reviewer. We measured the phosphorescence stability of the 7-MQ/BP under air, N₂ and O₂ atmospheres at room temperature. After exposure to different atmospheres, there was almost no change in phosphorescence intensity for the 7-MQ/BP (Supporting Figure I3), indicating excellent phosphorescence stability.

Supporting Figure I3. Phosphorescence intensity at 495 nm of the phosphor with a ratio of 1:1000 as a function of time under various atmosphere.

Comments #4: Metals or heavy atoms have not been involved in the materials, whereas the emission lifetimes are in millisecond scale. Is this caused by the structures of luminescent dopants, or benzophenone matrices, or some other reasons?

Response: Thanks for the comment of the reviewer. The molecular phosphorescence lifetime is closely related to the molecular structure and the environment around the molecules. From the molecular structure, the aromatic carbonyl group can boost the radiative transition rate for short-lived phosphorescence emission. At the same time, the carbonyl group environment provides rich n electrons for further shortening the phosphorescence lifetime (*Acc. Chem. Res.* 2022, 55, 1573; *Appl. Phys. Rev.* 2023, 10, 021313.). Meanwhile, the phosphorescence lifetime for 7-MQ in the BP and THF at 77 K is different. 7-MQ has a longer phosphorescence lifetime in the THF solution (Supplementary Fig. 22 and Fig. 27 in Supplementary information). From experimental data and theoretical calculation, we concluded that the BP matrix benefits short phosphorescence lifetime.

Summary: The contents of this study is well organized and this manuscript is easy to follow. I support the acceptance of this manuscript after rational revisions.

Response: We are grateful for the professional suggestions and highly positive comments from the reviewer.

Reviewer #1 (Remarks to the Author):

Although the current results are not satisfactory, it appears that the authors have sufficiently addressed the importance of research in this area. I recommend this manuscript to publish.

Reviewer #3 (Remarks to the Author):

The authors make a nice response to reviewer comments. In this version, the photophysical mechanism receives further supports from both experiments and analysis. The function of narrowband afterglow materials has been enriched from two additional aspects. These have been updated in the revised manuscript and supporting information. The reviewer recommends the acceptance of this revised manuscript.

Point-by-Point Response to Referees

Reviewer #1:

Although the current results are not satisfactory, it appears that the authors have sufficiently addressed the importance of research in this area. I recommend this manuscript to publish.

Response: Thanks to the reviewers for their approval of our current version of the article.

Reviewer #3:

The authors make a nice response to reviewer comments. In this version, the photophysical mechanism receives further supports from both experiments and analysis. The function of narrowband afterglow materials has been enriched from two additional aspects. These have been updated in the revised manuscript and supporting information. The reviewer recommends the acceptance of this revised manuscript.

Response: We appreciate the positive comments from the reviewer.